# Cholinergic Internal and Projection Systems of Hippocampus and Neocortex Critical for Early Spatial Memory Consolidation in Normal and Chronic Cerebral Hypoperfusion Conditions in Rats with Different Abilities to Consolidation: The Role of Cholinergic Interneurons of the Hippocampus

**DOI:** 10.3390/biomedicines10071532

**Published:** 2022-06-28

**Authors:** Elena I. Zakharova, Andrey T. Proshin, Mikhail Y. Monakov, Alexander M. Dudchenko

**Affiliations:** 1Laboratory of General Pathology of Cardiorespiratory System, Institute of General Pathology and Pathophysiology, Baltiyskaya, 8, 125315 Moscow, Russia; monakovm@mail.ru (M.Y.M.); amdudchenko@gmail.com (A.M.D.); 2Laboratory of Functional Neurochemistry, P.K. Anokhin Institute of Normal Physiology, Baltiyskaya, 8, 125315 Moscow, Russia; proshin_at@mail.ru

**Keywords:** spatial memory consolidation, learning stage 2s1, chronic cerebral hypoperfusion (2VO model), hippocampus, neocortex, cholinergic interneurons, cholinergic projection neurons, fractions of light and heavy synaptosomes, sub-fractions of synaptic membranes and synaptoplasm, membrane-bound and water-soluble choline acetyltransferase

## Abstract

The role of cholinergic projection systems of the neocortex and hippocampus in memory consolidation in healthy and neuropathological conditions has been subject to intensive research. On the contrary, the significance of cholinergic cortical and hippocampal interneurons in learning has hardly been studied. We aimed to evaluate the role of both cholinergic projection neurons and interneurons of the neocortex and hippocampus at an early stage of spatial memory consolidation (2s1) in normal and chronic brain hypoperfusion conditions. Control rats and rats subjected to permanent two-vessel occlusion were trained with the Morris water maze, and the activity of membrane-bound and water-soluble choline acetyltransferase was evaluated in the sub-fractions of ‘light’ and ‘heavy’ synaptosomes of the neocortex and hippocampus, in which the presynapses of cholinergic projections and interneurons, respectively, are concentrated. Animals were ranked into quartiles according to their performance on stage 2s1. We found: (1) quartile-dependent cholinergic composition of 2s1 function and dynamics of cholinergic synaptic plasticity under cerebral hypoperfusion; (2) cholinergic hippocampal interneurons are necessary for successful 2s1 consolidation; (3) cholinergic neocortical interneurons and projections can be critical for 2s1 consolidation in less learning rats. We conclude that targeted modulation of cholinergic synaptic activity in the hippocampus and neocortex can be effective in reversing the cognitive disturbance of cerebral hypoperfusion. We discuss the possible ways to restore the impaired spatial memory 2s1 in the presence of cerebral hypoperfusion.

## 1. Introduction

Neuronal, mediator-specific components of functional neuronal networks and the dynamics of their relationships underlie the study of brain functions in normal and pathological conditions [1,2,3]. This is one of the main modern trends in the search for understanding, maintaining, and correcting the cognitive status of the brain, including the functions of learning and memory.

The cholinergic afferents of the hippocampus and neocortex from the basal forebrain nuclei are widely known as modulators of neuronal activity in these structures and cognitive processes in animals and humans. However, studies on the role of cholinergic projection neurons in memory consolidation have produced ambiguous results, ranging from affirmation to denial of their global significance in the formation of a memory trace or assumptions about the participation of cholinergic systems only in events preceding or accompanying learning [4,5,6,7,8,9,10]. This discrepancy may be due in part to the variable methodological approaches. To us, however, it seems that the majority of studies have not taken into account two factors that affect the outcome of the experiment.

First, investigation of the functional properties of cholinergic interneurons of the neocortex and hippocampus along with the properties of cholinergic projections. For example, such studies have proven to be informative at the sub-synaptic level in biochemical research. In both the neocortex and hippocampus, cholinergic influence comes from two main sources, namely sub-cortical projection neurons and interneurons [11,12,13,14]. However, evidence for the involvement of cholinergic interneurons of the neocortex and hippocampus in cognitive functions is limited [1,15]. According to all these immunocytochemical, molecular genetic, and optogenetic data, the cholinergic synaptic pool in the neocortex and hippocampus is overwhelmingly represented by projection neurons, the power of which is ≥3 times greater than that of interneurons in the neocortex and even more in the hippocampus.

Our biochemical data on the ratio of cholinergic activity in the ‘light’ and ‘heavy’ fractions of synaptosomes in both brain structures are in good agreement with these data [16]. In addition, in our studies we have revealed functional differences (differences in response to the experimental effects) in cholinergic presynapses from light and heavy synaptosomes [16,17,18,19]. This approach has allowed for a high probability that interneuron presynapses are concentrated in the heavy synaptosomes and, thus, a differentiated study of cholinergic indicators in these synaptosomes makes it possible to identify the individual contribution of different populations of cholinergic neurons to the studied functions. Here it should be clarified that when modeling in vivo experiments, the studies were informative on sub-synaptic fractions of synaptic membranes and synaptoplasm, on which all our biochemical studies were performed.

Second, there is heterogeneity in the neuronal organisation of functions in individuals with different learning abilities. Thus, in experimental practice, the ratio of laboratory animals with different cognitive abilities in different experimental settings can vary greatly and thereby introduce variability into the results. Hence, the selection of individual cohorts of animals with different learning dynamics could be used as an experimental approach. The feasibility of this approach has been emphasised in some studies [17,20]. This endeavour could be especially fruitful to identify multiple and, possibly, non-linear mechanisms of involvement of various mediator components in memory formation, as well as to assess the adaptive and compensatory capabilities of mediator systems in CNS dysfunction.

Our first empirical experience of dividing a small group of intact rats according to learning dynamics revealed certain differences in the cholinergic organisation of spatial memory in rats with greater and less capability to learn [18]. The results of the analysis convinced us we were going in the correct direction. However, this approach requires an objective way to isolate cohorts of animals with a stable distribution profile under the conditions being studied.

In the present study, we used a statistical technique to divide trained rats into quartiles according to their ability to navigate in the Morris water maze (MWM), a behavioural task used to assess spatial learning and memory. To identify stable boundaries between quartiles of rats, we processed a statistically sufficient array of animals (see Section 2.6). The results represent the analysis of cholinergic correlates with memory consolidation in the normal brain and the two-vessel occlusion (2VO) model of chronic brain hypoperfusion at an early stage of the MWM, namely 2s1, which is the first trial on the second day of training, when skill automation is still absent [17].

We performed biochemical analysis on sub-fractions of synaptic membranes and synaptoplasm isolated from light and heavy synaptosomes of the neocortex and hippocampus. The light neocortical and hippocampal synaptosomal fractions (CC and HC, respectively) predominantly include the presynapses of projection neurons from the corresponding sub-cortical forebrain cholinergic nuclei. The heavy neocortical and hippocampal synaptosomal (CD and HD, respectively) fractions include predominantly presynapses of cholinergic neurons from another population, presumably interneurons [16]. In the sub-fractions of these synaptosomes, we evaluated the activity of membrane-bound (m) and soluble (s) choline acetyltransferase/acetyl-CoA:choline *O*-acetyl transferase (ChAT, EC 2.3.1.6) as a cholinergic marker. Moreover, in vivo experiments have convincingly shown that mChAT and sChAT can also indicate the functional state of cholinergic synapses and quantitative changes in the cholinergic synaptic pool [16,18,19,21].

We believed that the set of experimental approaches we have chosen will help to obtain new data on the complex problem of the cholinergic organisation of spatial memory both in normal conditions and in conditions of vascular pathology. Chronic cerebral hypoperfusion provokes delayed cognitive dysfunctions that cannot be corrected with the currently available treatments. Vascular dementia has long occupied a separate place among neurological diseases accompanied by cognitive deficits. According to modern concepts, all neurodegenerative diseases involve cerebral vascular pathology [22,23,24,25,26,27,28]. The experimental 2VO model has been used successfully in many studies to evaluate the mechanisms of delayed cognitive dysfunctions and to develop neuroprotective strategies in neurodegenerative diseases [22,29,30,31,32,33].

## 2. Materials and Methods

### 2.1. Animals and Ethical Approval

The behavioural experiments were carried out in male albino outbred laboratory rats aged 2.5–3.5 months (250–350 g). The animals came from the Light Mountains nursery (Moscow region, Russian Federation [RF]) and were kept in the vivarium of the Institute of General Pathology and Pathophysiology. All animal care and experimental procedures were carried out in accordance with the EU Directive 2010/63/EU. The number of experimental rats used and their suffering was minimised. The rats were housed in a temperature-controlled room (20–24 °C) and were maintained with a 12 h light-dark cycle. They were kept at 5–7 animals per rat cage of 40 × 60 cm in size, and had free access to food and water. All experimental protocols were approved by the Ethical Committee of the Institute of General Pathology and Pathophysiology (protocol #3 of 18 August 2021).

The rats were handled for at least two consecutive days prior to starting the behavioural experimental procedures. After the end of behavioural experiments, rats were decapitated with a guillotine, and samples were collected for biochemical experiments. The remaining animals were euthanised in CO_2_ inhalation using a euthanasia apparatus (Open Science, Krasnogorsk, Russia).

### 2.2. Chronic Cerebral Hypoperfusion

Chronic rat cerebral hypoperfusion was induced by permanent occlusion of the common carotid arteries by ligation (2VO model) [34]. The bilateral common carotid arteries were tied with silk threads while the rats were under an appropriate level of Nembutal anaesthesia (40 mg/kg body weight). The common carotid arteries were separated from the cervical sympathetic and vagal nerves through a ventral cervical incision. Sham-operated animals (control group) underwent the same surgical procedure with the exception of vascular ligation.

### 2.3. MWM

The MWM was performed following a published standard protocol [35]. The water maze was a circular pool (120 or 160 cm in diameter and 60 cm high) filled with milk-clouded water at 22–24 °C to a depth of 40 cm. A clear plexiglas platform (10 × 10 cm) was submerged 2 cm below the milk-clouded water surface in one of the quadrants of the pool. The platform position remained the same throughout the training period.

The rats were training on days 6–8 (2VO-7d group) or days 28–30 (2VO-1M group) after the surgery. At the beginning of all trials, the rats were placed in the pool at one of four starting positions. Trials lasted a maximum of 60 s; after the trial, the rat remained on the platform for 30 s. For rats that failed to find the platform within 60 s, the investigator carefully guided them towards the platform. Rats were trained during two, three, or four daily sessions (s) and each session consisted of four trials. In each trial, the latency to escape before reaching the platform was recorded as the task performance time (T, s).

### 2.4. Biochemical Analysis

The methods and procedures of preparation of sub-synaptic fractions from the fractions of synaptosomes of brain structures and ChAT activity determination were the same as described previously [36].

#### 2.4.1. Brain Tissue Preparation

All preparative procedures were performed at 2–4 °C. The hippocampus and neocortex were bilaterally separated from each brain and homogenised in an isosmotic solution. From each structure, synaptosomes were obtained from the rough mitochondrial fraction by centrifugation in a discontinuous sucrose gradient. Light synaptosomes were concentrated in the layers between 1.0 and 1.2 M sucrose and heavy synaptosomes between 1.2 and 1.4 M sucrose. Then, synaptosomes were precipitated and disrupted by the combined shock of suspending the synaptosomal pellets in a hypoosmotic solution and subsequent freeze-thaw exposure. The sub-fractions of synaptoplasm were obtained by centrifugation as supernatants from the disrupted synaptosomes. Then, the pellets of disrupted synaptosomes were also resuspended in the hypoosmotic solution and layered on a discontinued sucrose gradient. After centrifugation of both the light and heavy disrupted synaptosomes, the synaptic membrane sub-fractions were obtained in the layer between 0.6 and 1.2 M sucrose. To achieve an isosmotic condition, the sucrose suspensions of synaptic membrane sub-fractions were diluted with a solution containing 3 mM EDTA-Na_2_ and 9 mM Tris-HCl buffer, pH 7.4–7.5. All samples were stored at −80 °C until the day of the assay.

#### 2.4.2. ChAT Activity Assay

ChAT activity was determined by Fonnum’s radiometric method [37]. The final concentrations of the reaction were 0.2 mM acetyl CoA and (acetyl-1-^14^C)-CoA with SPA 5 mCi/mmol, 0.2 mM physostigmine salicylate, 10 mM choline chloride, 300 mM NaCl, 3 mM MgCl_2_, 0.5% Triton X-100, 0.5 mg/mL albumin from bull serum, 1 mM EDTA-Na_2_, 10 mM sodium phosphate buffer, pH 7.8, and 2.5–3.5 mg of sub-fraction samples. The reaction was incubated at 37 °C in a water shaker for 30–60 min, and the reaction was stopped by placing the mixture in an ice bath and with an excess solution of acetylcholine (0.2 mM) in the same sodium phosphate buffer. Then, a sodium tetraphenylborate solution in butyl acetate was added and the mix was shaken intensely in a shaker. The organic phase was carefully separated from the inorganic phase by centrifugation at low speed. The organic phase with [^14^C]-acetylcholine was moved into scintillation liquid for organic solutions and a radioactively of synthesised acetylcholine was quantified (DPM) with a beta counter.

#### 2.4.3. Reagents and Drugs

(Acetyl-1-^14^C)-CoA sodium salt was obtained from Amersham Pharmacia Bioscience (Amersham, Buckinghamshire, England); acetyl CoA sodium salt, physostigmine salicylate, tetraphenylborate sodium salt, choline chloride, naphthalene, tris(hydroxymethyl)aminomethane sodium salt and sucrose were obtained from Sigma-Aldrich (St. Louis, MO, USA); NaCl, MgCl_2_·H_2_O, Na_2_HPO_4_·2H_2_O, and ethylene glycol were obtained from Merck; toluol, butyl acetate, EDTA-Na_2_, PPO, POPOP, and some other reagents were obtained from REACHIM (Staraya Kupavna, Moscow region, Russia).

### 2.5. Experimental Protocol

Each rat was handled for at least two consecutive days before starting the experimental procedures.

The rats underwent surgical procedures and after 6–8 days (2VO-7d rats, group II, *n* = 19) or 1 month (2VO-1M rats, group III, *n* = 11) both 2VO and sham-operated (control rats, group I, *n* = 32) animals began training in the MWM. Experimental data were obtained in three different years, designated as experimental batches of rats ‘a, b, and c’.

The rats were trained for 2 days (experimental batche a, *n* = 14, *n* = 7 and *n* = 11 for control [I, a], 2VO-7d [II, a] and 2VO-1M [III, a] sub-groups, respectively), 3 days (experimental batche b, *n* = 11 and *n* = 7 for control [I, b] and 2VO-7d [II, b] sub-groups, respectively) or 4 days (experimental batche c, *n* = 7 and *n* = 5 for control [I, c] and 2VO-7d [II, c] sub-groups, respectively).

Two to four days after the end of training, the rats of all sub-groups were taken for the acute biochemical experiments. mChAT and sChAT activity was measured in sub-fractions, respectively, of synaptic membranes and synaptoplasm of light and heavy synaptosomes isolated from the neocortex and hippocampus. The terminology of De Robertis [38] was used to designate the light and heavy synaptosomes as C and D, respectively. These designations have been used in the Figures and Tables.

The data were obtained in a blinded manner. None of the experimenters knew the key characteristics of the tested rat. The experiments and animal preparation were performed by different experimenters.

In this work, all sub-groups were stratified into quartiles based on T values for the 2s1 stage (Table 1). In each quartile, the behavioural test results and the relationship of these indicators with ChAT activity in the synaptic populations of the neocortex and the hippocampus was analysed.

To conduct a comparative correlation analysis, identical sub-groups were gradually combined into the variational samples as experimental material was added.

### 2.6. Statistics

STATISTICA 8.0 was used for statistical analysis. The normality of the data was examined by using the Kolmogorov–Smirnov test (parameter d and *p* values). There were no deviations from normality. Microsoft Excel was used for correlation analysis, calculating Pearson correlation coefficients (*r*), and correcting for small samples (*n* ≤ 15) [39]. The differences were presented as individual points and the mean ± SEM. The results were subjected to the non-parametric Fisher’s exact test (FET criterion) and Wilcoxon–Mann–Whitney test (U criterion). The differences and correlations were considered significant at *p* < 0.05.

An important note on the correlation analysis of T-ChAT. The ‘+’ sign of the coefficient *r* means that the ChAT activity higher, the T value greater, that is, the worse the performance of behavioral task 2s1 (early memory consolidation). Accordingly, the sign ‘-’ has the opposite meaning: the higher the ChAT activity, the lower the T value, that is, the 2s1 consolidation is more successful. This contradicts the physiological significance of the dependence of T values on enzyme activity. Therefore, to display physiological significance, we presented the coefficient *r* of T-ChAT correlations with the opposite sign both in the text and in the illustrations. Thus, in the data on T-ChAT correlations, the sign ‘+’ means a positive relationship between ChAT activity and T values (the higher the enzyme activity, the more successful 2s1 consolidation), and the sign ‘−’ means the opposite.

Using Microsoft Excel, the data of each sub-group was divided into quartiles according to the ability to 2s1 performance. It was clear that the boundaries between quartiles are a species-specific attribute (specific for each learning model) and therefore should be more or less constant values, as well as the fact that only natural boundaries will make it possible to distinguish clusters of animals united by a common mechanism that determines their cognitive property. We identified fairly stable, presumably natural boundaries between quartiles (±0%–5%) for laboratory rats when processing data sets from ≥89 rats. Interval values of quartiles and boundaries between them at stages 2s1, 3s1, and 4s1 are presented in Table 2 for our two pools. According to the rules of statistics, the boundary T values were empirically included in the corresponding quartiles according to the correspondence of these variants to the correlation indicators of the main variation series.

Experimental data for each quartile was presented as a sequence of variational samples (Figure 1 and Figure 2, and Table 2, Table 3, Table 4 and Table 5). Experimental batche a made up the initial variational samples I, a, II, a (the sample consisted of the control sub-group I, a, and 2VO-7d sub-group II, a) and III, a (the sample consisted of the control sub-group I, a, and 2VO-1M sub-groups III, a, respectively).In the groups I and II, data from experimental batches b and c were added sequentially, namely: variational samples I, a + b (sub-groups I, a and I, b) and II, a + b (sample I, a + b and sub-groups II, a and II, b), and then variational samples I, a + b + c (sample I, a + b and sub-group I, c) and II, a + b + c (samples I, a +b + c and II, a + b and sub-group II, c). If the variational samples included only the variants of 2VO groups and did not include the variants of control group, then such samples had an additional Arabic numeral 1 in the designation (II, a, 1 or II, a + b, 1 or III, a, 1, etc.).

## 3. Results

### 3.1. Distribution of Control and 2VO Rats into Quartiles Based on MWM Performance

In the control (sham-operated) rats, the distribution of animals by quartiles according to the indicators of spatial memory consolidation on the second, third and fourth days of training were more or less uniform (Figure 3I). The quartile distribution of 2VO-7d rats (subjected to 2VO for seven days) contrasted sharply with the controls, with the majority of rats classified in the fourth quartile, indicating the inability to consolidate memory at all stages of learning (Figure 3II). For comparison, there were similar distribution patterns in both control and 2VO-7d rats based on the general data array we have obtained from this learning model (Figure 4I,II). A group of 2VO rats trained after a month of chronic hypoperfusion (2VO-1M, two days of training) demonstrated recovery of the 2s1 memory consolidation ability (Figure 3III).

### 3.2. ChAT Activity and Rat Spatial Learning Dynamics in Quartiles at the 2s1 Stage in Normal and Cerebral Hypoperfusion Conditions

In the control group in each studied sub-synaptic fraction, ChAT activity was similar in all quartiles. At the same time, the quartile correlation analysis of ChAT activity with task performance at the 2s1 stage (with T indicating the escape latency) revealed the uniqueness of the cholinergic composition of the 2s1 performance of each quartile in all the studied groups.

In order to demonstrate the stability of some correlations and instability of others within a quartile, we presented the fractional results of correlation analysis as the variational samples consisting of sub-groups of experimental batches a, and the gradual addition of the corresponding sub-groups of experimental batches b and c to them.

#### 3.2.1. First Quartile: Rats with High Memory Consolidation

In rats with high memory consolidation, mChAT activity of the HD fraction, alone or together with sChAT activity correlated positively with T values in all variational samples under all experimental conditions. For both the 2VO-7d and 2VO-1M groups, ChAT activity of this presynaptic population of the 1st quartile was resistant to hypoperfusion (Figure 1, 1st quartile; *r* values of the significant T-ChAT correlations are presented in Table 5, 1st quartile; *r* values of significant mChAT-sChAT correlations are presented in Table 3, 1st quartile).

Also, in controls, there were correlations between sChAT activity and T values in the CD or CC fraction in some variational samples (1, a or 1, a + b), and if they were observed, there were correlations between sChAT of both cortical fractions and mChAT activity in the HD fraction (Figure 1, scheme, and Table 4, 1st quartile, 1, a and 1, a + b).

In 2VO-7d rats, there was a conjugate decrease in both mChAT and sChAT activity in the CC fraction in the sample II, a + b by 53 ± 8% (*p* < 0.05) and 29 ± 4% (*p* < 0.05), respectively (Figure 1, graph, and Table 3, 1st quartile, II, a + b) and in the sample II, a + b + c by 47 ± 8% (*p* < 0.025) and 29 ± 9% (*p* < 0.025), respectively (Figure 1, graph, and Table 3, 1st quartile, II, a + b + c). In this fraction, a T–sChAT correlation was absent in 2VO-7d rats (Figure 1, 1st quartile, scheme, II, a + b and II, a + b + c).

In the CC fraction of 2VO-1M rats, mChAT and sChAT activity returned to control (Figure 1, 1st quartile, graph, III, a, 1 and III, a). In the variational sample III, a (control and 2VO rats) and not in III, a, 1 (2VO rats only), a T–sChAT correlation appeared in both CC and CD fractions and also T–mChAT correlation in the CD fractions (Figure 1, scheme, and Table 5, 1st quartile, III, a), and mChAT-sChAT correlations appeared (Figure 1, scheme, and Table 3, 1st quartile, III, a), and there were also cross-structural correlations with the HD fraction mChAT values (Figure 1, scheme, and Table 4, 1st quartile, III, a). It should be noted that the significant cross-correlations and mChAT-sChAT correlations also appeared in both cortical fractions of synaptosomes in sample III, a, 1 (Figure 1, scheme, and Table 3 and Table 4, 1st quartile, III, a, 1).

#### 3.2.2. Second Quartile: Rats with Medium-High Memory Consolidation

In control rats of the 2nd quartile, there was a T–mChAT and T–sChAT correlation in the CC fraction of all variational samples, and there was an mChAT–sChAT correlation in this faction of samples 1, a, 1, a + b and I, a + b + c (Figure 1 and Table 3 and Table 5, 2nd quartile, I, a, I, a + b and I, a + b + c).

Also in controls, there was a T–sChAT correlation in the HC fraction of sample I, a and a cross-structural correlation with sChAT activity in the HC and CC fractions (Figure 1 and Table 4 and Table 5, 2nd quartile, I, a).

For the 2VO-7d group, only one rat was in the 2nd quartile of the first two variational samples. The mChAT and sChAT activity in the CC fraction was higher than all control values by 41 and 38% in the II, a sample and 69 and 49% in the II, a + b sample, and there was a conjugated activation of mChAT and sChAT. However, there were no T–mChAT or T–sChAT correlations in the CC fraction as well as cross-structural correlation with the HC fraction (Figure 1, graphs and scheme, and Table 3, 2nd quartile, II, a and II, a + b).

In the variational sample II, a + b + c, there was a significant activation of mChAT by 58 ± 11% (*p* < 0.05) and, conversely, inhibition of sChAT by 20 ± 27% (*p* < 0.05) in the CC fraction. There was a T–sChAT correlation in this faction, as was observed in the corresponding control sample 1, a + b + c (Figure 1, graph and scheme, and Table 5, 2nd quartile, II, a + b + c).

For the 2VO-1M group, the ChAT activity did not differ from the control group, and a T–sChAT correlation was in the CC fraction. In the HC fraction, there was an uncoupled drop activity in mChAT by 66 ± 11% (*p* < 0.005) and sChAT by 41 ± 6% (*p* < 0.05), and a negative T–mChAT correlation appeared (Figure 1, graph and scheme, and Table 5, 2nd quartile, III, a).

Also, in the 2VO-1M, there was a negative T–mChAT correlation in the HD fraction (Figure 1, 2nd quartile, scheme). There was no T–ChAT correlation for either control or 2VO-7d rats in the HD fraction. Of note, sChAT activity in 2VO-7d rats was noticeably lower than all control values in this fraction by 79% in sample II, a and by 76% in sample II, a + b, and significantly lower by 41 ± 16% (*p* < 0.05) in sample II, a + b + c (Figure 1, 2nd quartile, graphs in dotted brackets). In sample III, a, there was a conjugate decrease in both mChAT activity by 42 ± 2% (*p* > 0.05) and sChAT by 54 ± 14% (*p* < 0.005) (Figure 1, graph and scheme, and Table 3, 2nd quartile, III, a).

In 2VO-1M rats, the mChAT activity in the HC fraction correlated with the sChAT activity in the CC fraction and mChAT activity in the HD fraction (Figure 1, scheme, and Table 4, 2nd quartile, III, a).

#### 3.2.3. Third Quartile: Rats with Middle-Low Memory Consolidation

In rats of the third quartile, there was a negative T–mChAT and/or T–sChAT correlation in the CD fraction in the control, 2VO-7d, and 2VO-1M groups, except for samples II, a, and II, a + b + c, 1 (Figure 2, scheme, and Table 5, 3rd quartile).

In this fraction, in the 2VO-7d group, mChAT and sChAT activity was reduced by 65% and 58, respectively, in the II, a sample and by 84 and 66%, respectively, in the II, a + b sample, and ChAT activity in this 2VO rat was lower than all control values. In the sample II, a + b + c, 1 and similar in sample II, a + b + c, a decrease in mChAT activity by 87 ± 9% was significant (*p* < 0.005) while a decrease in sChAT activity by 41 ± 1% was not significant (*p* > 0.05). In sample III, a, only one rat was in the 3rd quartile. The mChAT and sChAT activity was reduced by 65 and 58%, respectively, and their values were lower than all control values in the I, a sample. A drop in the mChAT and sChAT activity in the CD fraction was conjugated in all 2VO samples (Figure 2, graphs, and Table 3, 3rd quartile, all samples of groups II and III).

In the HD fraction of control rats, there was a negative T–mChAT correlation only in sample I, a + b (Figure 2, scheme, and Table 5, 3rd quartile). In this fraction, in 2VO-7d rats, mChAT activity was increased in all samples, by 75% in the II, a sample, by 130% in the II, a + b sample, and by 82 ± 8% in the II, a + b + c sample (*p* < 0.025) (Figure 2, 3rd quartile, graphs, all samples II), and there was no T–mChAT correlation. At 2VO-1M rats, there was normalised mChAT activity and restoration of the T–mChAT correlation (Figure 2, graph and scheme, and Table 5, 3rd quartile, III, a).

In the CC fraction of control rats, there was a negative T–mChAT correlation and a positive T–sChAT correlation in sample I, a (Figure 2, scheme, and Table 5, 3rd quartile). In this fraction, in 2VO-7d rats, mChAT and sChAT activity was reduced by 50 and 30%, respectively, for the II, a sample, and by 22 and 27%, respectively, for the II, a + b sample.

Both ChAT activity was lower than control values for the II, a sample and the sChAT activity for the II, a + b sample. sChAT activity was significantly reduced by 41 ± 17% in the II, a + b + c, 1/II, a + b + c samples (*p* < 0.05) (Figure 2, graph, 3rd quartile, II, a, II, a + b, II, a + b + c, 1 and II, a + b + c). The decrease in both ChAT activity was not conjugated. At 2VO-1M, in the CC fraction, the mChAT and sChAT activity was also reduced independently by 42% and 25%, respectively, and was lower than control values in sample I, a. There was a negative T–mChAT correlation in the CC fraction and a positive mChAT–sChAT correlation between both neocortical fractions (Figure 2, graph and scheme, and Table 1 and Table 3, 3rd quartile, III, a).

Also in the 2VO-7d group, there was a negative T–sChAT correlation in the HC fraction of II, a + b + c, 1 and II, a + b + c samples (Figure 2, scheme, and Table 5, 3rd quartile, II, a + b + c, 1 and II, a + b + c). There was no T–ChAT correlation in the control rats in this fraction.

#### 3.2.4. Fourth Quartile: Rats with Low Memory Consolidation

In the rats with low memory consolidation (4th quartile), mChAT and/or sChAT activity in the CD fraction was consistently positively associated with T values in the control, 2VO-7d, and 2VO-1M groups. ChAT activity of this presynaptic population of the 4th quartile was resistant to hypoperfusion, and a mChAT-sChAT correlation was in all variation samples of experimental batche a in this fraction (Figure 2, scheme, and Table 3 and Table 5, 4th quartile, all samples in groups I, II and III).

In addition to the aforementioned stable association, the positive T–sChAT correlation was observed in the HC fraction in control sample I, a. There was the cross-structural sChAT-sChAT correlation between HC and CD fractions ((Figure 2, scheme, and Table 4 and Table 5, 4th quartile, I, a). In the 2VO-7d group, in the HC fraction, there was a significant decrease of mChAT activity by 27 ± 8% (*p* < 0.05) in the II, a + b sample and by 22 ± 8% (*p* < 0.05) in the II, a + b + c, 1/II, a + b + c samples (, II, a + b + c, 1 and II, a + b + c). In the III, a sample of the 2VO-1M group, normalization of the mChAT activity (93 ± 12%, *p* > 0.05) and insignificant sChAT activation (132 ± 27%, *p* > 0.05) was observed in this fraction, but there was no restoration of T-sChAT association as in the control I, a sample (Figure 2, graph, 4th quartile, III, a).

In the 2VO-7d group, in the CC fraction, there was a significant activation of sChAT by 32 ± 17% (*p* < 0.05) and conjugated with sChAT insignificant mChAT activation by 8 ± 13% (*p* > 0.05) in the sample II, a + b + c, 1 or II, a + b + c, and there was a positive T–mChAT correlation in the same samples (Figure 2, graph and scheme, and Table 3 and Table 5, 4th quartile, II, a + b + c, 1 and II, a + b + c). In the III, a sample of 2VO-1M group, sChAT, and mChAT activity was identical to those in sample II, a + b + c, 1 or II, a + b + c in the CC fraction (33 ± 15% and 16 ± 16%, respectively). However, the sChAT (and mChAT) activity did not differ from control values (*p* > 0.05), and there was no a T-mChAT correlation in these samples (Figure 2, graph, 4th quartile, III, a).

Also, in the 2VO-7d group only, in the HD fraction, there was a negative T–sChAT correlation in the sample II, a + b + c, 1 (Figure 2, scheme, and Table 5, 4th quartile, II, a + b + c, 1)

In the 2VO-7d samples, there were the interfractional mChAT-mChAT correlation between CC and CD factions in the II, a + b + c, 1 and II, a + b + c samples and the cross-structural mChAT-sChAT correlations between CC and HD factions in the II, a + b + c, 1 sample (Figure 2, scheme, and Table 4, 4th quartile, II, a + b + c, 1 and II, a + b + c).

#### 3.2.5. Coherent 2VO-Mediated Changes in ChAT Activity in Different Synaptosome Fractions

In order to reveal a coherence of changes in ChAT activity in the different fractions of synaptosomes under 2VO conditions, a quartile correlation analysis was carried out in the variational samples in which these changes were observed in more than one fraction. These were all samples of the 2VO-7d group of rats in the second and third quartiles, as well as samples II, a + b + c, 1 and II, and a + b + c in the fourth quartile (Figure 1 and Figure 2, graphs).

In the second quartile, in all samples of the 2VO-7d group of rats, there was a stable negative correlation between an activation of mChAT in the CC fraction of the neocortex and a suppression of sChAT activity in the HD fraction of hippocampus (Table 2, 2nd quartile). Note that correlations between these fractions were not observed in control animals. At the same time, in the 2VO-7d samples, one value was enough for a correlation to arise.

In the third quartile, there was a negative correlation between mChAT activation in the HD fraction of hippocampus and suppression of mChAT and sChAT activity in the CD fraction of neocortex in sample II, a + b + c, 1, which included only 2VO-7d rats (Table 2, 3rd quartile).

Note that in the second and third quartiles, in sample III, a, the changes in ChAT activity were accompanied by the appearance of T-ChAT correlations in the corresponding fractions. Therefore, the coherence of changes in ChAT activity in these fractions was described above in Section 3.2.2 and Section 3.2.3 and was shown in the schemes (Figure 1 and Figure 2) and Table 4.

In the fourth quartile, there were no significant correlations between 2VO-induced changes in ChAT activity in the neocortical CC and hippocampal HC fractions, both in the samples II, a + b + c, 1 and II, a + b + c, and the sample III, a.

## 4. Discussion

### 4.1. Analysis of Spatial Learning Dynamics and General Characteristics of the Cholinergic Composition

By dividing rats into quartiles based on memory consolidation, we have revealed spatial learning dynamics and demonstrated a pronounced consolidation in 2VO-7d rats at all the studied stages (2s1, 3s1, and 4s1) and restoration of the ability to consolidate in 2VO-1M rats at the 2s1 stage (Figure 3). The phases of cognitive dysfunction and normalisation correspond with the results of a pilot magnetic resonance (MR) study. Specifically, the dynamics of venous blood perfusion was monitored in two 2VO-treated rats by using the susceptibility-weighted imaging (SWI) mode. An increase in cerebral hypoperfusion was found from 5 to 15 days of 2VO, the period within which 2VO-7d rats were trained. Moreover, there was a consistent weakening of hypoperfusion from 15 to 43 days of 2VO alongside the restoration of the blood supply to the brain due to the opening of collaterals (unpublished data). These results are consistent with those obtained by other researchers using a similar 2VO model [23,40,41,42].

Data on the marked impairment of long-term spatial memory at the critical period of seven days after 2VO at all the studied stages indicate the universality of the observed phenomenon. At the same time, normalisation of 2s1 performance at one month after 2VO has not been reported consistently in the literature. According to a number of authors, delayed death of hippocampal pyramidal neurons may occur in 2VO rats at or after 1M, a phenomenon that exacerbates the pathological consequences of hypoperfusion [42,43]. According to our biochemical data, despite restoration of 2s1 function, neuronal reorganization and probably synaptic degeneration continues at the 2VO-1M stage (Figure 1 and Figure 2, quartiles 2–4, III, a).

In the present work, we detected the stability of some cholinergic connections and the instability of others, findings consistent with the dynamics of 2VO-initiated plastic changes in the synaptic pool. Identification of these patterns required us to perform a detailed study. Hence, we conducted a fractional, step-by-step analysis of the results obtained in the control and 2VO rats, as far as our experimental data allowed (Figure 1 and Figure 2). With this endeavour, we have shown that the cholinergic composition of 2s1 performance, which is distinct for each quartile, requires cholinergic neurons, whose presynapses are concentrated in the CD and HD fractions. Moreover, within the quartiles, we have revealed non-random interfractional/interstructural connections between the key neurons involved in early consolidation and non-key (unstable) neuronal components. These connections reflect the consistency of action of different neuronal populations. This property underlies proper functioning of neural networks. As in normal conditions there are no interquartile differences in mChAT and sChAT activity, it follows that the individuality and features of the inclusion of cholinergic synaptic populations of 2s1 performance are the results of the organisation of neuronal networks of this function, distinct for each quartile.

### 4.2. Cholinergic Presynapses of Heavy Synaptosomes Are Key to 2s1 Performance

By employing a biochemical study, we found that in rats of the first quartile, 2s1 performance is stably and positively correlated with cholinergic presynapses of the HD fraction of synaptosomes (Figure 1. 1st quartile), and in rats of the fourth quartile with cholinergic presynapses of the CD fraction of synaptosomes (Figure 2, 4th quartile). These associations are also stable for the 2VO-7d and 2VO-1M groups. Thus, the inclusion of cholinergic presynapses of the HD fraction in the nervous network is necessary to manifest spatial navigation at the 2s1 stage, which is inherent in rats of the first quartile, as well as the cholinergic presynapses of the CD fraction for rats of the fourth quartile. In other words, the cholinergic presynapses of the HD and CD fractions are key to 2s1 performance in rats of the first and fourth quartiles, respectively.

It should be noted that the stable positive T–ChAT correlation in all variation samples (which is 6 and 9 samples for the first and fourth quartiles) inspires confidence in the method we used to divide the experimental rats into quartiles. The method objectively groups animals according to neuronal (and possibly other) mechanisms of the function under study.

In rats of both extreme quartiles (first and fourth), a stable correlation appeared with mChAT activity and, in certain experimental groups, with mChAT-associated sChAT activity (Table 5). Therefore, it is logical to assume that mChAT represents the same populations of presynapses in all variation samples. However, as a result of individual variability within the quartiles of synaptosomes in terms of the population composition of presynapses (the size of the synaptoplasm and active zones and, accordingly, the activity of the corresponding enzymes), a correlation with sChAT activity could manifest itself only in individuals in which the key population was quite representative.

The identity of cholinergic presynapses of the HD and CD fractions is unknown. We have reason to believe (and we have presented our arguments in Section 1) that in both the hippocampus and neocortex, these fractions are predominantly concentrated presynapses of cholinergic interneurons, the functional properties of which have not been well studied. The present study demonstrates a clear functional individuality of these presynaptic populations.

Some data allow us to put forward an alternative hypothesis, namely: sub-cortical cholinergic neurons that send projections to either of the two structures (the hippocampus or neocortex) are distributed topographically, which suggests their functional heterogeneity [44,45,46,47]. However, the complexity of the topographic picture of cholinergic innervation should be noted, and this question is still open, as well as the question of the distribution of presynapses of these neurons in the synaptosome fractions.

It is well known that the presynapses of projection neurons are predominantly small and medium in size and relatively light in density [11,13,48], the factors that determine their concentration in light synaptosomes [38]. This does not exclude the presence of a certain percentage of projection presynapses in heavy synaptosomes, especially presynapses with borderline sizes and densities. Nevertheless, the intrinsic nature of cholinergic neurons involved in the mechanism of long-term memory could explain the preservation of the ability to consolidate under conditions of lysis of cholinergic projections in the hippocampus, as well as many decades of controversy about the role of the hippocampal cholinergic system in memory consolidation. Of note, according to the present study of an early stage of spatial memory consolidation 2s1, presynapses of cholinergic projections in the hippocampus (HC fraction of synaptosomes) showed a stable T-ChAT correlation only in the experimental batche a (see next Section 4.3). This is a small part of rats from the total array of animals.

The participation of interneurons in the mechanism of information storage seems to be physiological, especially considering their localisation in the hippocampal CA1 and CA3 zones [15]. Cortical cholinergic interneurons and their role in learning and behaviour are beginning to be studied [1].

### 4.3. Cholinergic Presynapses Are Not Key for 2s1 Performance

In the extreme quartiles (first and fourth), there were inconsistent T–ChAT correlations with cholinergic presynapses of other synaptosomes; this finding also applied to the corresponding synaptosomes in the middle quartiles. For the first and fourth quartiles, these inconsistencies were different for the T–sChAT and/or T–mChAT correlations of presynapses of projection neurons of the neocortex and hippocampus (CC and HC fractions, respectively), as well as presynapses of all fractions of projective neurons and interneurons in the second and third quartiles (Figure 1 and Figure 2). We classified them as non-key because their absence was not decisive in the manifestation of the level of memory consolidation.

Of note, in intact rats of the second and third quartiles, 2s1 performance is stably correlated with the cholinergic presynapses of neocortical projective neurons (CC fraction) and interneurons (CD fraction), respectively (Figure 1 and Figure 2). Additional analysis about these fractions is set out in Section 4.5.

Also, the stable presence of some irregular T–ChAT correlations could be traced within the experiment batche. In the experiment batche a in the second quartile (Figure 1), in the cholinergic presynapses of hippocampal projective neurons (HC fraction), there was a positive T-sChAT correlation in the control sample I, a, which was preserved under 2VO conditions at stage 7d (sample II, a). At stage 1M (sample III, a), there was a significant drop in both mChAT and sChAT activity in this fraction, the T-ChAT correlation turned negative with mChAT and the association of the hippocampal projective neurons with memory consolidation was thus preserved. These data indicate the existence within a quartile of variations in the cholinergic composition of neuronal networks in rats with the same abilities. Hence, it could be possible to apply a multifactorial approach to potentiate memory consolidation under normal conditions or to correct it under 2VO conditions.

Two more findings from the 2VO-1M group indicate the non-randomness of irregular T-ChAT connections: (1) their restoration in the case of loss during normalisation of cholinergic activity (in the first, second, and third quartiles) and (2) restoration or the occurrence of new interstructural and interfractional correlations with the restored synaptic populations, including with the key populations of cholinergic interneurons (all quartiles). The functional significance of these irregular connections in memory consolidation is an open question.

### 4.4. Effects of 2VO on Synaptic ChAT Activity

It should be noted that in our pilot MR study of two rats, neither rat showed ischaemic brain injury. It has been repeatedly shown that cognitive deficit in the 2VO model develops against the background of the absence of ischaemic damage to brain structures, among other factors [23,29,33,41,49]. Moreover, all experiments were performed on surviving animals, which account for a maximum of 40% in 2VO simulations [42], including in our studies (unpublished data). That is, these are animals in which the effect of hypoperfusion on the brain was not lethal. Thus, synaptic plasticity under 2VO conditions may be the main cause of both pathogenic and adaptive-compensatory processes in the brain and, thus, the main cause of delayed cognitive dysfunctions, the cholinergic synaptic mechanisms of which and possible ways of their correction are analyzed here.

In 2VO rats, hypoperfusion initiated: (1) an associated decrease in mChAT and sChAT activity (in the first quartile CC fraction at stage seven days, the second quartile HD fraction at stage one month and the third quartile CD fraction at stage of both seven days and one month), which may reflect degeneration or elimination of the corresponding synaptic populations; (2) an unassociated decrease in mChAT and sChAT activity or one of them (in the second quartile HC fraction at stage one month and HD fraction at stage seven days, the third quartile CC fraction at stage seven days and one month and the fourth quartile HC fraction at stage seven days), which may indicate a decrease in the cholinergic effects of one or more independent synaptic populations due to their degeneration/elimination as well as inhibition/dysfunction of mediator activity in one of them (mChAT) and the synthesis of acetylcholine in the others (sChAT); (3) an associated increase in mChAT and sChAT activity (in the second and fourth quartiles CC fraction at stage seven days), which may reflect synaptogenesis (sprouting); and (4) activation of mChAT (in the second quartile CC fraction at stage seven days and the third quartile HD fraction at stage seven days) which suggests cholinergic activation of the mediator function (Figure 1 and Figure 2).

In models of neurodegenerative diseases, including 2VO, along with degeneration or elimination of fibres and synaptic contacts reflecting dysfunction [31,33,43,50,51], sprouting and mediator activation (hyperfunction) have been described [52,53,54,55]. Changes in mChAT and sChAT activity in response to in vivo influences are, as a rule, the equivalent of plastic changes in the cholinergic synaptic pool. We have repeatedly analysed and substantiated this issue [16,19,36,56,57].

According to the ChAT reaction, in the majority of cases, hypoperfusion provoked degeneration/elimination or inhibition/dysfunction of cholinergic synaptic populations and the disappearance of the corresponding presynapses from the neuronal network. Also, the connection with the function could also disappear with an increase in the power of cholinergic influences. In these cases, it is possible that reorganisation at the critical period of 2VO (seven days) could be aimed at allowing the individual to survive under adverse conditions [36]—for example, to enhance the efficiency of contacts with cerebral vessels or activation of inhibitory interneurons to suppress glutamate excitotoxicity [58]. At the same time, there may be another reason for an increase in the cholinergic pool. In the same model of chronic hypoperfusion, it was found that spatial learning and memory impairment, which began at seven days of 2VO, is accompanied by a long-term increase in the number of silent synaptic connections and a decrease in the number of functional synapses in the CAI area of the hippocampus [33]. The authors hypothesize that such changes in silent synapses may underlie the cellular basis of cognitive impairment induced by hypoperfusion.

### 4.5. Possible Pathways for the Transition of 2VO-7d Rats to the Lower Quartiles

Under the influence of hypoperfusion at the critical time of 7 days, the 2s1 performance of about 75% of the experimental rats was classified in the fourth quartile (Table 1). Obviously, most of these animals belonged to higher quartiles under normal conditions. It is also likely in the 2VO-7d group that not only the fourth quartile animals, but also the middle quartiles, included a mixture of individuals both retained intact 2s1 consolidation abilities and those with dysfunction from the higher quartiles. This made it difficult to analyze the dynamics of reorganization of the cholinergic synaptic pool under 2VO conditions.

Nevertheless, we were able to trace some possible pathways of cholinergic reorganisation that promote the transition to the lower and lowest level of memory consolidation abilities at the 2s1 stage.

**In the first quartile** (Figure 1, 1st quartile), the following parallels can be traced: at stage seven days, a decrease in the pool, probably degeneration/elimination of cholinergic presynapses of cortical projections (CC fraction) and disappearance of both neocortical projections and interneurons (CD fraction) from the 2s1 neuronal network; at stage 1 month, restoration of quantitative indicators of projection presynapses and connection with the function of both projections and interneurons, including interstructural relationships with key interneurons (HD fraction).

Further, we believe that the reason for the transition of rats of the first quartile at stage seven days of 2VO to lower ones is a disorder in the key population of cholinergic presynapses of the HD fraction.

So, **in the second quartile** (Figure 1, 2nd quartile) in the norm, cholinergic presynapses of the HD fraction did not participate in 2s1 consolidation; at stage seven days in this fraction, there was a deep inhibition of acetylcholine synthesis (a drop in sChAT activity) or degeneration/elimination of relatively large presynapses. This correlated with cortical mChAT activation (CC fraction) across all 2VO-7d samples (Table 4, 2nd quartile). The composition of ChAT activity and cholinergic organization of 2s1 function at stage one month did not match the changes at stage seven days. At the same time, it can be explained by a complex of changes in 2VO rats of the second quartile with different native neuronal organization:

(1) Coherent changes in the sChAT-mChAT activity in the HD-CC fractions occurred in first quartile rats due to a disorder in the key population of hippocampal presynapses that led to a decrease in abilities to 2s1 consolidation to the level of the second quartile. These individuals, when restored, at stage one month, returned to the first quartile.

(2) In parallel with cholinergic activation in the sub-sample of rats from the first quartile, in the CC fraction projective presynapses, inhibition of acetylcholine synthesis (a drop in the sChAT activity) occurred in the native population of rats of the second quartile. These changes did not disrupt the coherence of changes in the sChAT-mChAT activity in the HD-CC fractions in samples II, a, and II, a + b (in which, apparently, rats from the first quartile dominated) and manifested themselves in a significant and mChAT-independent decrease in the sChAT activity in sample II, a + b + c (in which the predominance of the first quartile was, apparently, not sufficiently pronounced).

At stage 1 month, the sChAT activity returned to normal, and neocortical projective presynapses were restored in the neural network of the second quartile. It should be noted here that the T-sChAT correlation in the CC fraction was also observed in sample II, a + b + c (apparently, as a result of a sufficient representation of native rats of the second quartile in this sample). It is then likely that in native rats of the second quartile, the 2s1 function did not lose the CC fraction at stage seven days and, therefore, neocortical cholinergic projections are key for this quartile.

(3) Inhibition of synaptic function and acetylcholine synthesis in the cholinergic projections of HC fraction (presumably in different presynapse populations of this fraction, since the decrease in mChAT and sChAT activity were not conjugated) also apparently occurred in the native population of rats of the second quartile, selectively in experimental batche a, as we already reported above, in Section 4.3. Obviously, these inhibitory processes were initiated after a critical period of seven days and led to the formation of a new, negative T-mChAT association at stage one month.

(4) Degeneration of cholinergic presynapses in the HD fraction (associated drop in mChAT-sChAT activity) at stage one month apparently also occurred in the native rat population of the second quartile after stage seven days and also led to the formation of a new, negative T-mChAT association at stage one month.

As a result, at stage one month of 2VO, the ability to consolidate 2s1 in the second quartile was realized through a reorganized neural network, with new cholinergic hippocampal components and cross-correlations with them of restored (or key) neocortical cholinergic projections.

**In the third quartile** (Figure 2, 3rd quartile) at stage seven days, along with its own reorganization, a set of changes apparently also occurred in the brain of rats with a different native neuronal organization:

(1) In the cholinergic presynapses of cortical projections (CC fraction), there was a suppression of acetylcholine synthesis (decrease in sChAT activity) and the disappearance of this fraction from the 2s1 neuronal network. At stage one month, there was a restoration of the connection between the CC fraction and function, as well as the emergence of interstructural relationships with the cholinergic presynapses of the CD fraction. At the same time, ChAT activity was reduced.

Here it should be noted that only one rat entered the third quartile at the 2VO-1M stage, and, accordingly, control values prevail in the correlation analysis of sample III, a. Therefore, the data for this sample are preliminary.

(2) There was sustained activation of cholinergic presynapses (mChAT activation) in the HD fraction and degeneration/elimination of cholinergic presynapses in the CD fraction (decrease in mChAT activity associated with sChAT activity). Coherence was found between changes in mChAT activity in the HD fraction with mChAT and sChAT activity in the CD fraction only in sample II, a + b + c, 1, which included the 2VO-7d rats without control rats (Table 4, 3rd quartile). It points to hypoperfusion as the initiator of these relationships.

Changes in the HD fraction could occur in the brains of native rats of the third quartile, since the activation of choliergic presynapses of the HD fraction eliminated their inclusion in the 2s1 function and the normalization of cholinergic activity in the fraction at stage 1 month coincided with the restoration of these presynapses to the functional network. However, another version seems to us more probable.

Persistent degeneration in the synaptic pool of the CD fraction is not associated with irregular T-ChAT correlation in the presynapses of this fraction in 2VO rats. Considering this, as well as the coherence of changes in the CD and HD fractions, we conclude that coherent changes are inherent in the brains of rats from another quartile, namely, from the first quartile. Because only the first quartile in the control showed the same interstructural relationships (compared with Figure 1, first quartile, sample I, a). At stage one month, as in the second quartile, the brains of these rats likely normalized 2VO-7d disorders and could return to the first quartile. At least at this stage, neither such pronounced changes were observed as at the critical period of seven days, nor their coherence.

And then it is possible that in the native rats of the third quartile, the 2s1 function did not lose the CD fraction at stage seven days, and the cortical cholinergic interneurons are key for this quartile.

**In the fourth quartile** at stage seven days, two-thirds of the rats entered from higher quartiles. Therefore, it was especially difficult to identify a source of cholinergic reorganization under conditions of hypoperfusion in this quartile. We present some of our assumptions (Figure 2, fourth quartile):

(1) There was synaptogenesis of neocortical projections in the samples II, a + b = c, 1 and II, a + b = c (sChAT activation conjugated with mChAT activity in the CC fraction), association with 2s1 function in both samples, which was not observed in the control samples, as well as interfractional mChAT-mChAT association with the HC fraction, which is key for the fourth quartile. It is possible that the potentiation of neocortical projections was a summation multidirectional response to hypoperfusion in individuals from all quartiles that were in the fourth quartile at stage seven days. At the same time, only in the fourth quartile was a relationship observed between the CC and HC fractions, in which mChAT positively correlated with 2s1 performance. Therefore, we suggest that the T-mChAT correlation was inherent in rats of the fourth quartile and emerged as a new connection in the 2s1 neuronal network due to the reorganization of cholinergic presynapses of neocortical projections.

(2) There was inhibition of synaptic function in the cholinergic projections of the hippocampus (a decrease in mChAT activity in the HC fraction) in the samples II, a + b, II, a + b = c, 1, and II, a + b = c, similar to the second quartile at the stage one month. At the same time, there was no T-ChAT correlation in the HC fraction, as in the second quartile, as well as the restoration of T-ChAT correlation with normalization of ChAT activity at the stage one month, which was observed in the control sample I, a in rats of the fourth quartile.

The apparent absence of T-ChAT correlation of hippocampal projections under conditions of hypoperfusion may be the result of a similar cholinergic response in rats of the second and fourth quartiles and an inclusion of projections in the 2s1 functional network with a negative effect on 2s1 performance in rats from the second quartile and, conversely, with a positive effect on 2s1 performance in rats of the fourth quartile.

It is also possible that in rats of one of the quartiles, cholinergic reorganization in the projection system of the hippocampus specifically for the level of the fourth quartile eliminated them from the functional 2s1 consolidation network.

In any case, the fourth quartile of 2VO rats includes at least two sub-samples of rats, since changes in the CC and HC fractions did not conjugate in the samples III, a + b + c, 1, and III, a + b + c.

(3) Pays attention that the predominance of non-native rats in the fourth quartile did not disrupt the association of the CD fraction with the 2s1 function. We assume the transition to the fourth quartile of a sufficient number of rats from the first quartile with a positive T-ChAT correlation in the functional 2s1 network (as in the control group of the first quartile I, a) and its preservation against the background of disturbances in the HD fraction (for example, as it manifested itself in the second quartile at the stage seven days).

(4) In the interneurons of the hippocampus (fraction HD), changes in the activity of ChAT could not be manifested, apparently due to multidirectional deviations in the activity of enzymes in sub-samples of rats from the first quartile, similar to those observed in the middle quartiles.

Apparently, for the same reason, the degeneration/elimination of synapses of the HD fraction, similarly observed in the second quartile at the stage one month, did not manifest itself. At the same time, we assume that changes of this type in hippocampal interneurons in a sub-sample of rats from the second quartile accompanied their transition to the fourth quartile. The presence of these rats in the fourth quartile appeared on day seven in sample II, a + b + c, 1 as a negative T-sChAT correlation in the HD fraction and an interstructural association of HD and CC fractions similar to those in the second quartile at stage one month. A disorder in this sub-sample of cholinergic function or network connections of the presynapses of the CC fraction could be a reason for this transition, if our assumption is correct and the neocortical projections are key for the second quartile.

Obviously, the fourth quartile included individuals from the higher quartiles in which the neuronal reorganization of the 2s1 function developed according to the most dramatic scenario. According to our data, this can be indicated by events in the HC and HD fractions, which in certain sub-samples of rats in the fourth quartile were observed earlier than in the middle quartiles at the critical stage 2VO-7d. It is also possible that the changes in other fractions were more profound, i.e., the differences with the middle quartiles were quantitative, or these cohorts of rats had critical changes in the other neuronal populations of functional network 2s1.

In general, based on the specifics of the quartile cholinergic organization of 2s1 performance, there does not seem to be a universal cholinergic regulation of spatial learning in normal conditions or its maintenance/restoration under hypoperfusion conditions.

Under normal conditions, we do not see obvious ways in cholinergic organization for potentiation of 2s1 execution to move into a higher quartile. Apparently, in the 2s1 functional network, a component of another mediator specificity is such binder. For example, this role can perform the dopaminergic projections to the hippocampus, critical for the successful consolidation of spatial memory in the early stages of learning [59]. According to our recent study, the activity of tyrosine hydroxylase in the hippocampus showed both broad associations with hippocampal and neocortical ChAT activity and association with 2s1 performance across all quartiles, the rapidity of which was directly proportional to enzyme activity [60]. This is an important and interesting problem. Rats that are incapable of the consolidation of spatial memory during the course of learning (4 days), after a two-week break show the high abilities for memory consolidation [17], the mechanisms of which are unknown.

Under 2VO conditions, reorganization of the cholinergic synaptic pool, individual for each quartile, provided information on possible ways to restore the ability to consolidate 2s1. It should be noted the high plasticity of hippocampal cholinergic interneurons and the prospects for using this plasticity. Correction of impaired function by modulating the efficiency of these interneurons could bring learning dynamics to a high level. At stage seven days, this can be a directed effect (1) on the cholinergic interneurons of hippocampus or projections of the neocortex in the sub-sample of animals that fell into the second quartile, (2) on the cholinergic interneurons of the hippocampus or neocortex in the sub-sample of animals that fell into the third quartile. It follows from our data that it is necessary to know the native abilities of an individual in order to choose a recipe for correcting impaired function.

Also, of note is that the fourth quartile is the most representative at the critical stage of hypoperfusion of seven days and, thus, the most popular in the study of mechanisms of vascular dementia. Unfortunately, the complex composition of the fourth quartile and the low information content of obtained data made our conclusions on this quartile the most rough. The study of this issue may require additional methodological approaches.

It should be taken into account that in individuals with chronic hypoperfusion, the neuronal organisation of functions would not be completely identical to that in the intact brain: the brain is being ‘rewired’ in accordance with the changed environmental conditions [61]. This is also evidenced by our experimental data. Modulation of new neuronal elements and connections of the functional network may also be effective in chronic cerebral hypoperfusion.

Finally, the distribution of early consolidation abilities occurred in our experimental sample well consistent with the distribution in the actual array of 2VO rats (Figure 3II and Figure 4II). Therefore, we conclude that the pronounced shift towards cognitive deficits is purposeful. It is known that the neurons involved in cognitive functions are the most energy-consuming [58] and it is possible that any reorganisation of neuronal networks leading to cognitive deficit is aimed at maintaining the viability of the organism in the conditions of brain pathology. Thereby, compensatory-restorative support should be carried out at the stage of hypoperfusion when it is safe for the organism.

## 5. Conclusions

Neuroplasticity, the ability of the brain to change and adapt as a result of the interaction of the body with the environment, underlies the restoration of impaired functions, primarily learning and memory under pathological conditions affecting the brain [61]. Synaptic plasticity is the most immediate and dynamic way of adapting, compensating, and restoring impaired functions. Clearly, understanding dysfunction requires knowledge of the natural biological basis of function in the intact brain.

The quartile method for data analysis is used in a number of psychological and clinical studies. At the same time, our work seems to be the first or one of the first in the neurobiological experiments. In the present study, we employed the quartile method to combine experimental rats into cohorts according to the 2s1 performance of the MWM, a spatial learning and memory task. We obtained new data on the synaptic cholinergic components of the neuronal organisation of this function and made the following findings:Quartile-dependent cholinergic composition of the studied function and pathways of synaptic cholinergic plasticity seven days and one month after onset of chronic cerebral hypoperfusion;Key and non-key cholinergic elements of the neural network of the function;Poorly studied synapses of cholinergic interneurons of the hippocampus and neocortex (presynapses of heavy synaptosomes) play a key role in 2s1 consolidation;Cholinergic hippocampal interneurons are critical for successful 2s1 consolidation and correction of dysfunction provoked by cerebral hypoperfusion;Cholinergic neocortical interneurons and projections can be critical for 2s1 function in normal and chronic brain hypoperfusion conditions in rats with middle and low memory consolidation abilities;Multifactorial nature of potentiation of memory consolidation in the norm and correction under cerebral hypoperfusion conditions.

We put forward a working hypothesis that delayed cognitive dysfunction is perhaps the most common natural way of brain adaptation in conditions of chronic cerebral hypoperfusion, aimed at maintaining the energy resources necessary for the viability of the organism. Therefore, under hypoperfusion, it is necessary to study the temporal dynamics of the physiological status and compensatory processes to determine the phase that is optimal for the restoration of cognitive functions. It also seems promising to study the dynamics of restoration or reorganisation of the quartile-dependent receptor signalling of the memory consolidation function under conditions of hypoperfusion. This knowledge may reveal the way to correct cognitive dysfunctions.

## Figures and Tables

**Figure 1 biomedicines-10-01532-f001:**
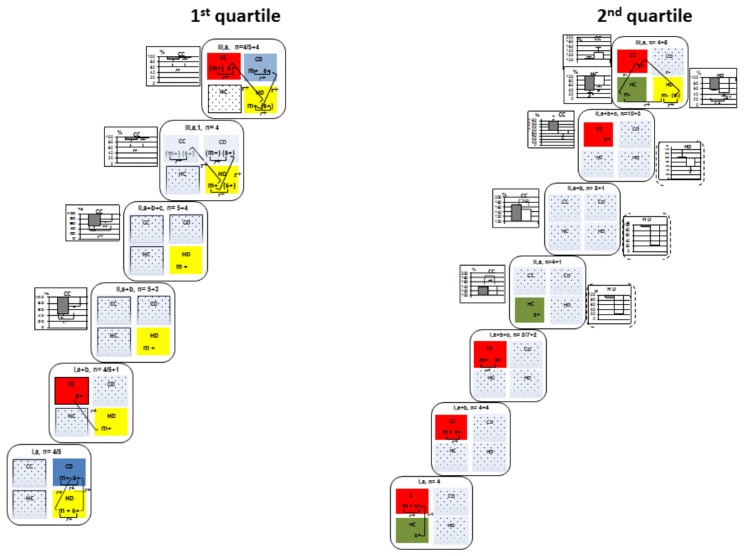
Scheme of the cholinergic composition of spatial memory consolidation at the 2s1 stage in rats of the 1st quartile (high memory consolidation) and 2nd quartile (middle-high memory consolidation). Large ovals represent the variational samples. (a), (b) and (c), experimental batches that made up the variation samples of rats (see Section 2.6); (I), (II) and (III), control, 2VO-7d and 2VO-1M groups of rats, respectively; n, number of variants of each sample. The samples (II) and (III), data include the corresponding control and 2VO sub-groups. The samples with an added Arabic numeral (1) in the designations, data include the corresponding 2VO sub-groups only. Squares in ovals represent synaptosomes: (CC) and (CD) are light and heavy neocortical synaptosomes, respectively, and (HC) and (HD) are light and heavy hippocampal synaptosomes, respectively. The colour (red, blue, green, yellow) indicates the fractions in which a significant T–ChAT correlation was observed. In this fractions, (m), membrane-bound ChAT (mChAT); (s), water-soluble ChAT (sChAT); (m+) or (s+), positive T–ChAT correlation; (m−) and (s−), negative T–ChAT correlation, (m) or (s) enclosed in oval brackets indicate medium strength T–ChAT correlations (*r* > 0.4, *p* > 0.05), Pearson’s *r*-test. See Table 5 for the correlation values. The sign (+) and (−) in T-ChAT correlation indicates the physiological meaning of the relationship between T values and ChAT activity, as opposed to the mathematical meaning in a T–ChAT correlation (see Section 2.6). In the variational samples (II) and (III), changes in ChAT activity are reflected in the graphs located next to the corresponding synaptosomes in the sample. In the graphs, the *y*-axis represents per cent change (individual points and the mean ± SEM) in ChAT activity compared with control (set at 100%); dark bars are mChAT activity and light bars are sChAT activity; (*) *p* < 0.05, (**) *p* < 0.025, (3*) *p* < 0.01, and (4*) *p* < 0.005, Fisher’s exact test and Wilcoxon–Mann–Whitney test. In the schemes and graphs, the angle brackets connecting (m) and (s) within a fraction show the significant mChAT–sChAT correlation, Pearson’s *r*-test (see Table 3 for the correlation values); the angle brackets connecting (m) or (s) in different factions indicate the interfractional significant ChAT–ChAT correlation, Pearson’s *r*-test (see Table 4 for the correlation values). Next to brackets, (*r*+) indicates a positive ChAT-ChAT correlation, while (*r*−) represents a negative ChAT-ChAT correlation.

**Figure 2 biomedicines-10-01532-f002:**
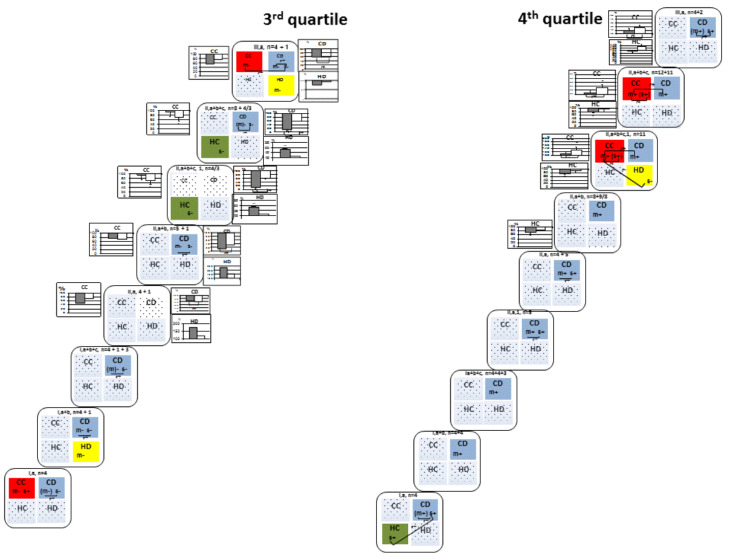
Scheme of the cholinergic composition of spatial memory consolidation at the 2s1 stage in rats of the 3rd (medium-low memory consolidation) and 4th quartiles (low memory consolidation). In the graphs, (**) *p* < 0.025, Fisher’s exact test. The rest of the designations are as in Figure 1.

**Figure 3 biomedicines-10-01532-f003:**
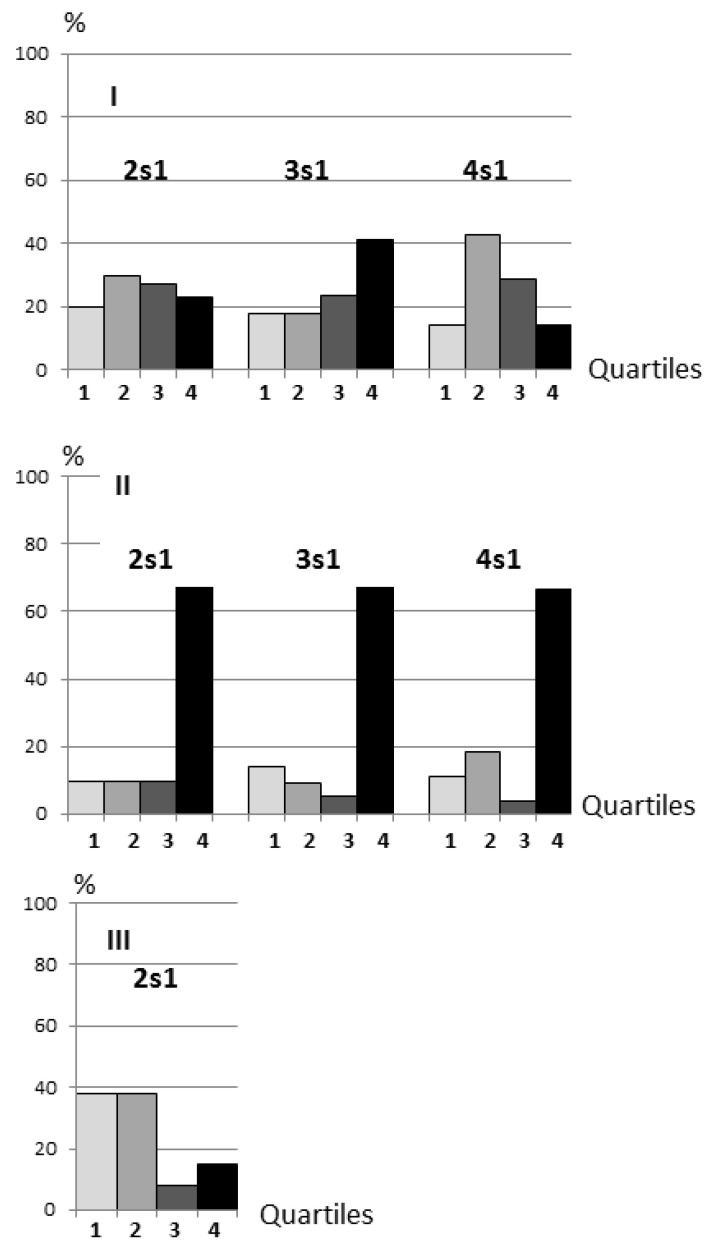
Per cent distribution of rats in quartiles based on their ability to consolidate spatial memory in the experimental array of our biochemical studies. (**I**), control group; (**II**), group 2VO-7d, chronic cerebral hypoperfusion for 7 days; and (**III**), group 2VO-1M, chronic cerebral hypoperfusion for 1 month. 2s1, 3s1 and 4s1, the first test (memory consolidation index) on the second, third and fourth day of training, respectively. The total number of trained rats in each group (**I**–**III**) at each training stage (2s1, 3s1 and 4s1) is 100%. (1), (2), (3), (4) on the *x*-axis: (1), 1st quartile (high memory consolidation); (2), 2nd quartile (middle-high memory consolidation); (3), 3rd quartile (medium-low memory consolidation); (4), 4th quartile (low memory consolidation). The total number of trained rats: *n* = 32, *n* = 17 and *n* = 7 for the 2s1, 3s1 and 4s1 training stages, respectively, in group (**I**); *n* = 24, *n* = 25 and *n* = 13 for the 2s1, 3s1 and 4s1 training stages, respectively, in group (**II**); *n* = 11 for the 2s1 training stage in group (**III**).

**Figure 4 biomedicines-10-01532-f004:**
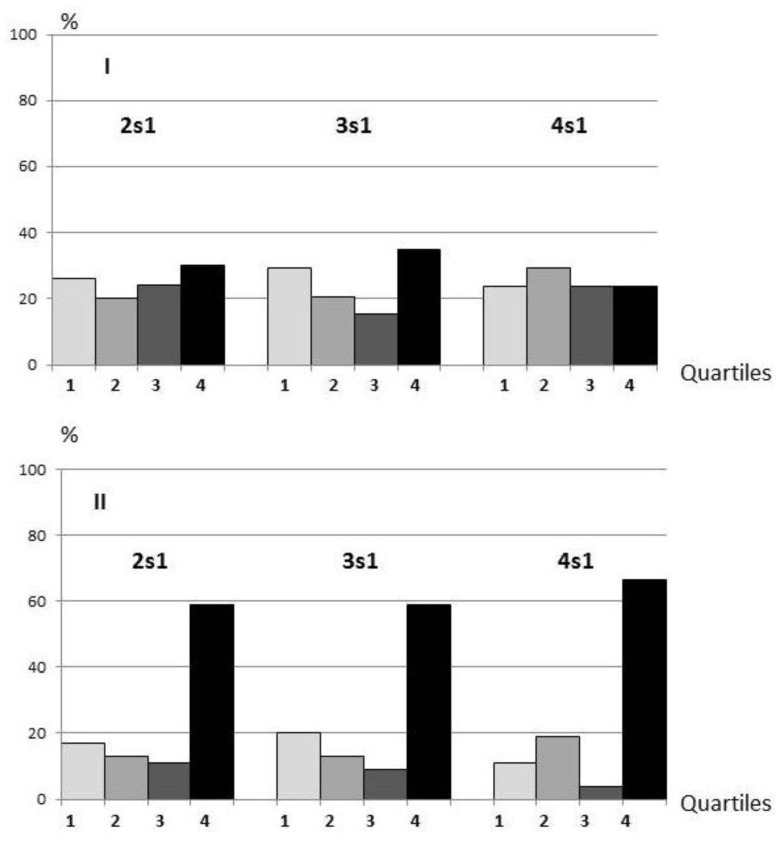
Per cent distribution of rats in quartiles based on their spatial memory consolidation, with statistical significance in norm (control group) for determining the stable boundaries between quartiles (see Section 2.6). Designations are the same as in Figure 3. The total number of trained rats: *n* = 236, *n* = 241 and *n* = 96 for the 2s1, 3s1 and 4s1 training stages, respectively, in group (**I**), and *n* = 46, *n* = 46 and *n* = 27, respectively, for the 2s1, 3s1 and 4s1 training stages, respectively, in group (**II**).

**Table 1 biomedicines-10-01532-t001:** Natural (stable) boundaries between quartiles by indicators of the escape latency to the hidden platform (T, s) in laboratory rats in the first trial to solve the task on the second day of training (2s1) using the Morris water maze with a pool (A) 120 cm and (B) 160 cm in diameter.

WLM, d, cm	Escape Latency Test of Hidden Platform Achievement (T, s)	Quartile 1 High Capable Rats	Quartile 2 Middle-High Capable Rats	Quartile 3 Middle-Low Capable Rats	Quartile 4 Low Capable Rats
A	2s1, *n* = 123	≤12	≥12–≤27	≥27–≤48–47	≥48–47
A	3s1, *n* = 123	≤7	≥7–≤13	≥13–≤27	≥27
B	2s1, *n* = 113	≤18	≥18–≤35–38	≥35–38–≤59–60	≥59–60
B	3s1, *n* = 118	≤8–9	≥8–9–≤21–23	≥21–23–≤41–42	≥41–42
B	4s1, *n* = 96	≤7	≥7–≤14	≥14–≤27	≥27

**Table 2 biomedicines-10-01532-t002:** Significant ChAT–ChAT Activity Correlations between Hypoperfusion-mediated Changes in Neocortical and Hippocampal Synaptosomes for Rats of each Quartile.

2nd quartile, to Figure 1, 2nd quartile					
**Fraction**	**Pirson’** **Test**	**II, a**	**II, a + b**	**II, a + b + c, 1**	**II, a + b + c**
		**HD, sChAT**	**HD, sChAT**	**HD, sChAT**	**HD, sChAT**
CC, mChAT	*r*	−0.920	−0.705		−0.724
	*p*, *n*	*p* < 0.05, *n* = 5	*p* < 0.05, *n* = 9		*p* < 0.01, *n* = 12
CC, sChAT	*r*				
	*p*, *n*				
3rd quartile, to Figure 2, 3rd quartile					
**Fraction**	**Pirson’** **Test**	**II, a**	**II, a + b**	**II, a + b + c, 1**	**II, a + b + c**
		**HD, mChAT**	**HD, mChAT**	**HD, mChAT**	**HD, mChAT**
CC, mChAT	*r*				
	*p*, *n*				
CC, sChAT	*r*				
	*p*, *n*				
CD, mChAT	*r*			−0.997	
	*p*, *n*			*p* < 0.01, *n* = 4	
CD, sChAT	*r*			−0.963	
	*p*, *n*			*p* < 0.05, *n* = 4	

The table includes only samples with significant interstructural correlations identified in the second and third quartiles. *r*−, negative ChAT–ChAT correlation. Other designations are the same as in Table 1.

**Table 3 biomedicines-10-01532-t003:** **Significant mChAT–sChAT Activity Correlations in the Neocortical/Hippocampal Synaptosomes for Rats of each Quartile.** For the correlations, *r*+ indicates a positive mChAT–sChAT correlation; *r*−, negative mChAT–sChAT correlation. Other designations are the same as in Table 5.

1st quartile, to Figure 1, 1st quartile								
**Fraction**	**Pirson’** **Test**	**I, a**		**II, a + b**	**III, a, 1**		**III, a**	
CC	*r*			+0.868	+0.999		+0.713	
	*p*, *n*			*p* < 0.01, *n* = 9	*p* < 0.001, *n* = 4		*p* < 0.05, *n* = 8	
CD	*r*	+0.917			+0.995		+0.934	
	*p*, *n*	*p* < 0.05, *n* = 5			*p* < 0.02, *n* = 4		*p* < 0.001, *n* = 9	
HC	*r*							
	*p*, *n*							
HD	*r*	+0.987			+0.962		+0.823	
	*P*, *n*	*p* < 0.02, *n* = 4			*p* < 0.05, *n* = 4		*p* < 0.01, *n* = 9	
2nd quartile, to Figure 1, 2nd quartile								
**Fraction**	**Pirson’** **Test**	**I, a**		**I, a + b**	**II, a**	**II, a + b**	**III, a**	
CC	*r*	+0.998		+0.894	+0.997	+0.911		
	*p*, *n*	*p* < 0.02, *n* = 4		*p* < 0.01, *n* = 7	*p* < 0.01, *n* = 4	*p* < 0.01, *n* = 8		
CD	*r*							
	*p*, *n*							
HC	*r*	+0.999						
	*p*, *n*	*p* < 0.001, *n* = 4						
HD	*r*						0.974	
	*p*. *n*						*p* < 0.001, *n* = 9	
3rd quartile, to Figure 2, 3rd quartile								
**Fraction**	**Pirson’** **Test**	**I, a**	**I, a + b**	**I, a + b + c**	**II, a + b**	**II,** **a + b + c, 1**	**II,** **a + b + c**	**III, a**
CC	*r*					−0.964		
	*p*, *n*					*p* < 0.05, *n* = 4		
CD	*r*	+0.992	+0.956	+0.832	+0.943	+0.953	+0.694	+0.994
	*p*, *n*	*p* < 0.01, *n* = 4	*p* < 0.02, *n* = 4	*p* < 0.05, *n* = 7	*p* < 0.02, *n* = 5	*p* < 0.05, *n* = 4	*p* < 0.02, *n* = 11	*p* < 0.001, *n* = 5
HC	*r*							
	*p*, *n*							
HD	*r*							
	*p*. *n*							
4th quartile, to Figure 2, 4th quartile								
**Fraction**	**Pirson’** **Test**	**I, a**	**II, a, 1**	**II, a**	**II, a + b + c, 1**	**II, a + b + c**		**III, a**
CC	*r*					+0.916	+0.754	+0.969
	*p*, *n*				*p* < 0.001, *n* = 11	*p* < 0.001, *n* = 23		*p* < 0.02, *n* = 6
CD	*r*	+0.995	+0.978	+0.918				+0.931
	*p*, *n*	*p* < 0.01, *n* = 4	*p* < 0.01, *n* = 5	*p* < 0.001, *n* = 9				*p* < 0.02, *n* = 6
HC	*r*							
	*p*, *n*							
HD	*r*							
	*p*. *n*							

**Table 4 biomedicines-10-01532-t004:** Significant ChAT–ChAT Activity Correlations between Different Brain Structures and Different Fractions within the Same Structure in Neocortical and Hippocampal Synaptosomes for Rats of each Quartile.

1st quartile, to Figure 1, 1st quartile					
**Fraction**	**Pirson’** **Test**	**I, a**	**I, a + b**	**III, a**	**III, a**
		**CD, sChAT**	**CC, sChAT**	**CC, sChAT**	**CD, sChAT**
HD, mChAT	*r*	+0.980	+0.911	+0.755	+0.715
	*p*, *n*	*p* < 0.01, *n* = 5	*p* < 0.05, *n* = 5	*p* < 0.05, *n* = 8	*p* < 0.05, *n* = 8
HD, sChAT	*r*	+0.965			
	*p*, *n*	*p* < 0.05, *n* = 4			
2nd quartile, to Figure 1, 2nd quartile					
**Fraction**	**Pirson’** **Test**	**I, a**	**I, a + b**	**III, a**	**III, a**
		**CC, sChAT**		**CC, sChAT**	**HD, mChAT**
HC, mChAT	*r*			−0.813	+0.755
	*p*, *n*			*p* < 0.01, *n* = 9	*p* < 0.05, *n* = 7
HC, sChAT	*r*	+0.974			
	*p*, *n*	*p* < 0.05, *n* = 4			
3rd quartile, to Figure 2, 3rd quartile					
**Fraction**	**Pirson’** **Test**	**I, a**	**I, a + b**	**III, a**	**III, a**
				**CC, mChAT**	
CD, mChAT	*r*				
	*p*, *n*				
CD, sChAT	*r*			+0.988	
	*p*, *n*			*p* < 0.02, *n* = 4	
4th quartile, to Figure 2, 4th quartile					
**Fraction**	**Pirson’** **Test**	**I, a**	**II, a + b + c, 1**	**II, a + b + c, 1**	**II, a + b + c**
		**CD, sChAT**	**CD, mChAT**	**HD, sChAT**	**CD, mChAT**
HC, mChAT	*r*				
	*p*, *n*				
HC, sChAT	*r*	+0.953			
	*p*, *n*	*p* < 0.05, *n* = 4			
CC, mChAT	*r*		+0.657	−0.725	+0.591
	*p*, *n*		*p* < 0.05, *n* = 11	*p* < 0.02, *n* = 11	*p* < 0.01, *n* = 23
CC, sChAT	*r*				
	*p*, *n*				

The table includes only samples with significant interfractional/interstructural correlations. *r*+ values, positive ChAT–ChAT correlation; *r*− values, negative ChAT–ChAT correlation. Other designations are the same as in Table 1.

**Table 5 biomedicines-10-01532-t005:** Significant Pearson Correlation Coefficients (*r*) of T (Escape Latency) with mChAT and/or sChAT Activity of Neocortical and Hippocampal Synaptosomes for Rats of each Quartile.

1st quartile, rats with high memory consolidation abilities 2s1, to Figure 1, 1st quartile										
**Fraction**	**Pirson’** **Test**	**I, a**		**I, a + b**			**II, a + b**	**III, a, 1**		**III, a**
CC, T-mChAT	*r*							(+0.509)		(+0.430)
	*p*, *n*							*p* > 0.05, *n* = 4		*p* > 0.05, *n* = 8
CC, T-sChAT	*r*			+0.970				(+0.663)		+0.729
	*p*, *n*			*p* < 0.01, *n* = 5				*p* > 0.05, *n* = 4		*p* < 0.05, *n* = 8
CD, T-mChAT	*r*							(+0.665)		+0.663
	*p*, *n*							*p* > 0.05, *n* = 4		*p* < 0.05, *n* = 10
CD, T-sChAT	*r*	+0.905						(+0.627)		+0.712
	*p*, *n*	*p* < 0.02, *n* = 5						*p* > 0.05, *n* = 4		*p* < 0.05, *n* = 9
HC, T-mChAT	*r*									
	*p*, *n*									
HC, T-sChAT	*r*									
	*p*, *n*									
HD, T-mChAT	*r*	+0.957		+0.942			+0.987	+0.963		+0.688
	*p*. *n*	*p* < 0.02, *n* = 5		*p* < 0.01, *n* = 6			*p* < 0.001, *n* = 9	*p* < 0.05, *n* = 4		*p* < 0.05, *n* = 9
HD, T-sChAT	*r*	+0.983						(+0.889)		(+0.577)
	*p*, *n*	*p* < 0.02, *n* = 4						*p* > 0.05, *n* = 4		*p* > 0.05, *n* = 9
2nd quartile, rats with medium-high memory consolidation abilities 2s1, to Figure 1, 2nd quartile										
**Fraction**	**Pirson’** **Test**	**I, a**		**I, a + b**		**I, a + b + c**	**II, a**	**II, a + b + c**		**III, a**
CC, T-mChAT	*r*	(+0.804)		+0.839						
	*p*, *n*	*p* > 0.51, *n* = 4		*p* < 0.02, *n* = 7						
CC, T-sChAT	*r*	+0.976		+0.776		+0.775		+0.692		+0.679
	*p*, *n*	*p* < 0.05, *n* = 4		*p* < 0.05, *n* = 8		*p* < 0.01, *n* = 10		*p* < 0.01, *n* = 13		*p* < 0.05, *n* = 9
CD, T-mChAT	*r*									
	*p*, *n*									
CD, T-sChAT	*r*									
	*p*, *n*									
HC, T-mChAT	*r*									−0.714
	*p*, *n*									*p* < 0.05, *n* = 10
HC, T-sChAT	*r*	+0.998					+0.986			
	*p*, *n*	*p* < 0.01, *n* = 4					*p* < 0.01, *n* = 5			
HD, T-mChAT	*r*									−0.754
	*p*. *n*									*p* < 0.05, *n* = 8
HD, T-sChAT	*r*									(−0.483)
	*p*, *n*									*p* > 0.05, *n* = 8
3rd quartile, rats with medium-low memory consolidation abilities 2s1, to Figure 2, 3rd quartile										
**Fraction**	**Pirson’** **Test**	**I, a**		**I, a + b**		**I, a + b + c**	**II, a**	**II, a + b**	**II, a + b + c**	**III, a**
CC, T-mChAT	*r*	−0.979								−0.991
	*p*, *n*	*p* < 0.05, *n* = 4								*p* < 0.001, *n* = 5
CC, T-sChAT	*r*	+0.982								
	*p*, *n*	*p* < 0.02, *n* = 4								
CD, T-mChAT	*r*	(−0, 763)		−0.972		(−0, 563)		−0.935		−0.966
	*p*, *n*	*p* > 0.05, *n* = 4		*p* < 0.05, *n* = 4		*p* > 0.05, *n* = 7		*p* < 0.02, *n* = 5		*p* < 0.01, *n* = 5
CD, T-sChAT	*r*	−0, 972		−0.918		−0.849		−0.889	−0.619	−0.903
	*p*, *n*	*p* < 0.05, *n* = 4		*p* < 0.05, *n* = 5		*p* < 0.01, *n* = 8		*p* < 0.05, *n* = 5	*p* < 0.05, *n* = 11	*p* < 0.05, *n* = 5
HC, T-mChAT	*r*									
	*p*, *n*									
HC, T-sChAT	*r*									
	*p*, *n*									
HD, T-mChAT	*r*			−0.978						−0.950
	*p*. *n*			*p* < 0.05, *n* = 4						*p* < 0.02, *n* = 5
HD, T-sChAT	*r*									
	*p*, *n*									
4th quartile, rats with low memory consolidation abilities 2s1, to Figure 2, 4th quartile										
**Fraction**	**Pirson’** **Test**	**I, a**	**I, a + b**	**I, a + b + c**	**II, a, 1**	**II, a**	**II, a + b**	**II, a + b + c, 1**	**II, a + b + c**	**III, a**
CC, T-mChAT	*r*							+0.603	+0.495	
	*p*, *n*							*p* < 0.05, *n* = 11	*p* < 0.002, *n* = 23	
CC, T-sChAT	*r*									
	*p*, *n*									
CD, T-mChAT	*r*	+0.983	+0.874	+0.839	+0.972	+0.706	+0.760	+0.688	+0.732	(+0.693)
	*p*, *n*	*p* < 0.02, *n* = 4	*p* < 0.01, *n* = 8	*p* < 0.01, *n* = 11	*p* < 0.01, *n* = 5	*p* < 0.05, *n* = 9	*p* < 0.001, *n* = 17	*p* < 0.02, *n* = 11	*p* < 0.001, *n* = 22	*p* > 0.05, *n* = 6
CD, T-sChAT	*r*	+0.995			+0.896	+0.901				+0.849
	*p*, *n*	*p* < 0.01, *n* = 4			*p* < 0.05, *n* = 5	*p* < 0.001, *n* = 8				*p* < 0.05, *n* = 6
HC, T-mChAT	*r*									
	*p*, *n*									
HC, T-sChAT	*r*									
	*p*, *n*									
HD, T-mChAT	*r*									
	*p*, *n*									
HD, T-sChAT	*r*							−0.621		
	*p*, *n*							*p* < 0.05, *n* = 11		

The variational samples and designations are the same as in Figure 1: (I), (II), and (III), control, 2VO-7d and 2VO-1M groups of rats; (a), (b) and (c), experimental batches that made up the variation samples of rats; the samples (II) and (III) include the values of corresponding control and 2VO rats; the samples with an added Arabic numeral (1) in the designations represent data for corresponding 2VO rats only; (CC), light cortical synaptosomal fraction; (CD), heavy cortical synaptosomal fraction; (HC), hippocampal light synaptosomal fraction; (HD), hippocampal heavy synaptosomal fraction. (T), escape latency; (mChAT), membrane-bound choline acetyltransferase; (sChAT), water-soluble acetylcholine transferase. (*r*), Pearson’s test criterion. For the correlations, *r*+ values indicates a positive (T–ChAT) correlation; *r*− values, a negative (T–ChAT) correlation. Recall that the sign (+) and (−) indicates the physiological meaning of the relationship between T values and ChAT activity, as opposed to the mathematical meaning in a T–ChAT correlation (see Section 2.6). The T–ChAT *r* values in parentheses are paired with significant associated mChAT–sChAT correlations (see Table 3).

## Data Availability

The data presented in this study are available on request from the corresponding author.

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
