# Peer review of "Cholinergic Internal and Projection Systems of Hippocampus and Neocortex Critical for Early Spatial Memory Consolidation in Normal and Chronic Cerebral Hypoperfusion Conditions in Rats with Different Abilities to Consolidation: The Role of Cholinergic Interneurons of the Hippocampus"

_biomedicines, 2022, doi:10.3390/biomedicines10071532_

Round 1
Reviewer 1 Report
This manuscript describes the analysis of cognitive decline in a chronic ischemia model in the hippocampus and neocortex of the hippocampus. The concept and idea of the manuscript is very good, but these data are too inadequate in explanation and results to interpret the results. I understand what you are trying to say about Figure 1, 2 and 3, but there is no explanation and the readers could not understand them.
Author Response
General address to Reviewers
Dear Reviewers,
We would like to thank you for agreeing to review our manuscript (Manuscript ID-1: ijms-1580952).
My co-authors and especially first author want to sincerely thank you for the objective and useful assessment of our manuscript and also apologize for submitting it for consideration in such an indistinct and fuzzy, “raw” redaction of our material. This is my main fault, the first author. The only thing that may partly excuse me is that the “idea” also seemed interesting and obvious to me, but the material is very complex, made in several comparison plans, and I did not take into account the complexity of its perception.
Also, please excuse us for the late reply. For the objective reasons, we could not immediately proceed to the second editing of the manuscript. The editorial board of the journal met us halfway and allowed the necessary delay.
In the second reduction, the manuscript was seriously revised. We also supplemented our data with experimental material that had not yet been obtained earlier. The following most significant changes have been made:
- In the designations of all Figures and Tables (the unnecessary was removed and replaced with the necessary, which should simplify perception);
- Additional variational samples in Figures 3 and 4, in accordance with additional experimental material, which statistically confirmed or refined data with single deviations at the 2VO-7d stage in samples in which n=1 for 2VO rats;
- Relevant additional statistical data in Tables 1-3;
- Additional table (Table 4 in second reduction of the manuscript);
- Extensive details in the description of the results (section 2 "Results"), analysis of the obtained data (section 3 "Discussion") and methods (section 4 "Materials and Methods").
All changes and additions are marked with a green marker.
Editing of English language was done by PRS (certificate available). We made all corrections and additions to the manuscript after proofreading. We tried to stick to the style and grammar given by the proofreader.
Respectfully,
Dr. Elena I. Zakharova,
Dr. Alexander M. Dudchenko,
Dr. Andrey T. Proshin,
Dr. Mishail Yu. Monakov

Reviewer 2 Report
In the current manuscript authors investigated the role of cholinergic projection in the memory consolidation by focusing on two brain subregions (hippocampus and neocortex). It is an interesting topic but there are several issues that are needed to be considered:
- Title of the manuscript is long and confusing.
- In general manuscript needs editing of English language.
- Abstract is confusing and is not clear what they found and what is the conclusion of the study.
- It is not clear how tissue processing is done to collect the tissue?
- What was the reason of selecting 6 days and one month after surgery to do behavioral experiment?
- How Chronic Cerebral Hypoperfusion model is established? If this model is already established , references are needed to be added,
- It is not clear why authors specifically selected Chronic Cerebral Hypoperfusion model for investigating the role of cholinergic projection in the memory consolidation? Why they did not select for example animal models of depression which memory dysfunction is one of the important symptoms?
- Which hippocampus is used for study? Right or left side?
- If the non parametric test was used to do comparison, correlation analysis should be done by using non paramedic test like Spearman's correlation?
- Authors have different groups while they used Wilcoxon–Mann–Whitney test which is suitable for comparison between 2 groups.
Author Response

(The authors gave the same response as above.)

Round 2
Reviewer 1 Report
I judge the manuscript to be acceptable.
Reviewer 2 Report
Authors responded to the comments in a satisfactory manner.